# Fungicidal Activity of Volatile Organic Compounds Emitted by *Burkholderia gladioli* Strain BBB-01

**DOI:** 10.3390/molecules26030745

**Published:** 2021-01-31

**Authors:** Ying-Tong Lin, Cheng-Cheng Lee, Wei-Ming Leu, Je-Jia Wu, Yu-Cheng Huang, Menghsiao Meng

**Affiliations:** 1Graduate Institute of Biotechnology, National Chung Hsing University, 250 Kuo-Kuang Rd., Taichung 40227, Taiwan; yongdandandan1211@gmail.com (Y.-T.L.); lichanjan@gmail.com (C.-C.L.); wmleu@nchu.edu.tw (W.-M.L.); yeasre08756@gmail.com (Y.-C.H.); 2Ph.D. Program in Microbial Genomics, National Chung Hsing University, 250 Kuo-Kuang Rd., Taichung 40227, Taiwan; lavida75818@gmail.com

**Keywords:** *Burkholderia gladioli*, *Magnaporthe oryzae*, volatile organic compounds, dimethyl disulfide, 2,5-dimethylfuran, fumigant

## Abstract

A *Burkholderia gladioli* strain, named BBB-01, was isolated from rice shoots based on the confrontation plate assay activity against several plant pathogenic fungi. The genome of this bacterial strain consists of two circular chromosomes and one plasmid with 8,201,484 base pairs in total. Pangenome analysis of 23 *B. gladioli* strains suggests that *B. gladioli* BBB-01 has the closest evolutionary relationship to *B. gladioli* pv. *gladioli* and *B. gladioli* pv. *agaricicola*. *B. gladioli* BBB-01 emitted dimethyl disulfide and 2,5-dimethylfuran when it was cultivated in lysogeny broth and potato dextrose broth, respectively. Dimethyl disulfide is a well-known pesticide, while the bioactivity of 2,5-dimethylfuran has not been reported. In this study, the inhibition activity of the vapor of these two compounds was examined against phytopathogenic fungi, including *Magnaporthe oryzae*, *Gibberella fujikuroi*, *Sarocladium oryzae*, *Phellinus noxius* and *Colletotrichum*
*fructicola*, and human pathogen *Candida albicans*. In general, 2,5-dimethylfuran is more potent than dimethyl disulfide in suppressing the growth of the tested fungi, suggesting that 2,5-dimethylfuran is a potential fumigant to control plant fungal disease.

## 1. Introduction

Over 2 million tons of chemical pesticides are used each year worldwide to enhance agriculture production, which otherwise would be reduced due to competitive exclusion by weeds or predation by various pests including insect herbivores, nematodes and pathogenic microorganisms [1]. Nonetheless, the heavy use of chemical pesticides has had negative impacts on both human health and environmental sustainability due to the lack of selective toxicity and their recalcitrant nature. Microbial antagonists may provide an alternative or a supplement to chemical pesticides for the protection of agricultural plants from pests through diverse mechanisms such as production of anti-pathogen substances and niche competition. Anti-pathogen substances produced by microorganisms could be liquid diffusible or volatile compounds. The former includes antibiotics, toxins, bio-surfactants, cell wall disruption enzymes, etc., while the latter are diverse chemical metabolites with low molecular mass, low boiling points, and high vapor pressures. Recently, volatile organic compounds (VOCs) emitted by microorganisms have gained increasing attention due to their importance in both basic science and the application potential in pest control [2,3].

Approximately 2000 VOCs, emitted from almost 1000 bacterial and fungal species, are archived in mVOC database 2.0 (http://bioinformatics.charite.de/mvoc/). They are categorized based on chemical structures and microbial emitters/receivers [4]. In view of chemical structures, microbial VOCs may belong to hydrocarbons, alcohols, ketones, ethers, esters, terpenes, heteroatom-containing compounds, and others. Microbial VOCs play diverse biological functions in the ecosystem; they may exert antagonistic or symbiotic effects among inter- and intra-kingdom species, modulate plant growth, and even condition plants to ward off pathogens. A couple of recent reviews are excellent reading sources to update the study progress in microbial VOCs [2,3].

Crops are constantly attacked by pathogenic fungi. For example, rice, a crucial food staple in Asia, is susceptible to many fungi. Infection of *Magnaporthe oryzae*, a causative agent of rice blast disease, may significantly devastate rice yields, causing from 20 to 30% losses each year [5,6]. Besides, *Gibberella fujikuroi* and *Sarocladium oryzae* are two other common pathogenic fungi of rice in Taiwan, which cause the bakanae disease [7] and the sheath rot disease [8], respectively. The long-term goal of this study is to seek potential biocontrol agents, by which the use of chemical pesticides in agriculture practice, particularly in the rice field, could be reduced in the future. *Burkholderia gladioli*, an aerobic gram-negative bacterium, often displays antagonistic interactions with other microbes in the environment. Here, we describe a *B. gladioli* strain, named strain BBB-01 hereafter, which was isolated from rice shoots. It exhibits a strong and broad antagonistic activity against many plant pathogenic fungi based on the confrontation plate assay. Moreover, *B. gladioli* BBB-01 emits a vast fraction of dimethyl disulfide or 2,5-dimethylfuran, depending on the culture condition. According to the mVOC database, dimethyl disulfide could be emitted by a broad spectrum of microorganisms, and it has been used as an active ingredient in commercial pesticides. By contrast, only a couple of fungi was reported to emit 2,5-dimethylfuran, without a notion about the compound’s bioactivity [9]. This report presents the antifungal activity of 2,5-dimethylfuran via fumigation for the first time. In addition, the efficacy of 2,5-dimethylfuran vapor in suppressing pathogenic fungi is compared with that of dimethyl disulfide.

## 2. Results

### 2.1. Bacteria Exhibiting Antifungal Activity

A previous collection of plant growth-promoting microorganisms, isolated from a variety of environmental sites/habitats, e.g., urban soils, farmland soils, plant leaves, and insect guts, was screened for antifungal activity by the standard plate confrontation assay in this study. The challenged pathogens used in the assay included M. oryzae, G. fujikuroi, S. oryzae and Phellinus noxius. The first three are pathogens of rice, while P. noxius infects a broad range of horticultural plants such as tea and fruit trees. The screen revealed that microbial isolates numbered #2, #3, #4, #5, #6, #20, and #21 exhibited significantly antagonistic effects on the growth of *M. oryzae* and *P. noxius* (Figure 1A). Among them, the isolate #2, i.e., *Burkholderia gladioli* BBB-01, was particularly noted due to its strong inhibition activity against all the four fungi.

To see whether the morphology of *M. oryzae* was changed upon confrontation by *B. gladioli* BBB-01, the mycelia at the front line facing the *Burkholderia* colony or the medium control was taken and observed under a scanning electron microscope (SEM) (Figure 1B). The mycelia confronted by *B. gladioli* BBB-01 showed fragmented appearances, while that taken from the control region was intact. This morphology change clearly indicates that *B. gladioli* BBB-01 truly killed the fungus rather than slowed down the fungal growth.

### 2.2. Phylogenetic Relationship of B. gladioli BBB-01 with Its Kin

*B. gladioli* is present in diverse habitats and possibly associated with humans, animals and plants. Strains of this species could be a plant pathogen, a lung pathogen in cystic fibrosis patients, or a contaminant in fermented foods producing deadly bongkrekic acid. On the other hand, some strains of *B. gladioli* may benefit its eukaryotic hosts by providing protective secondary metabolites such as antibiotics and other bioactive substances. Given its importance in the biologically relevant context, to date, hundreds of *B. gladioli* strains have been isolated and the genomic information is accessible in public databases for many of them. Recently, a pangenomic analysis of 206 *B. gladioli* strains was performed, separating these strains into five evolutionary clads [10]. All the known bongkrekic acid-producing strains and those containing biosynthetic gene cluster (BGC) for bongkrekic acid synthesis are in either clade 1A, 1B, or 1C. Strains of *B. gladioli* pv. *allicola* dominant clade 2, while *B. gladioli* pv. *gladioli* and *B. gladioli* pv. *agaricicola* dominant clade 3.

The genome of *B. gladioli* BBB-01 was sequenced using Illumina MiSeq System and Nanopore technology. The joining efforts resulted in two circular chromosomes with 4,180,205 and 3,913,767 bp, respectively, and one plasmid with 107,512 bp (Figure 2). The whole genome, 68.2% in GC content, contains 7665 protein- and 57 tRNA-coding genes. In this study, the phylogenetic relationship of *B. gladioli* BBB-01 with its kin was established. The sequencing reads of 22 other *B. gladioli* strains, selected from each clade of the previous classification by Jones et al. [10], were downloaded and individually *de novo* assembled. The strain name, accession number, and classification information of these 22 selected strains are listed in Table 1. Genome annotation and pangenome analysis were performed using the software Prokka (v1.14.5) [11] and Roary (v3.11.2) [12], respectively. The pangenome of these 23 *B. gladioli* strains consist of 15,046 genes, within which 4141 are core genes. The core genes were concatenated and used to conduct a multiple nucleotide sequence alignment. Accordingly, a maximum-likelihood tree was constructed using RAxML (v. 8.2.12) [13] in the GTR-GAMMA model with 1000 rapid bootstraps. This small-scale phylogeny has *B. gladioli* BBB-01 positioned in clade 3 (Figure 3), while the rest 22 strains were grouped identically as the previous classification [10]. Annotation of the pangenome in this study also indicates that only the *B. gladioli* strains belonging to the clade 1 contain BGCs analogous to BGC0000173 in MIBiG database, which denotes the production potential of bongkrekic acid. Therefore, *B. gladioli* BBB-01 is not, theoretically, a bongkrekic acid-producing strain.

Besides bongkrekic acid, *B. gladioli* strains could produce a variety of secondary metabolites according to the relevant genes classified in the COG category Q (Secondary metabolites biosynthesis, transport and catabolism). Specifically, there are 223 genes belonging to the category Q in *B. gladioli* BBB-01 and 458 genes in the pangenome of these 23 *B. gladioli* strains. Thus, distribution of each of the genes in the pangenome was analyzed by heatmap clustering. The resulting dendrogram reveals several hierarchical clusters depicted on the top of Figure 4. The dendrogram also indicates that the variation in the genetic contents involved in secondary metabolite synthesis, transport, and catabolism is more or less consistent with the clade classification in the core gene-based phylogeny.

### 2.3. B. gladioli BBB-01 Being Capable of Producing Antifungal Vapor

The confrontation plate assay showed antifungal activity of *B. gladioli* BBB-01 through direct interaction (Figure 1). It was interesting to know whether *B. gladioli* BBB-01 emits volatile compounds to suppress the fungal growth as well. The question was answered by conducting a simple test using a sealed compartment of two Petri dishes, in which the downward plate contained *M. oryzae* grown on PDA and the upward plate contained the bacterium grown on PDA, LBA, or KBA. The growth of *M. oryzae* was examined after a 10-day incubation at 28 °C. In comparison with the control, *B. gladioli* BBB-01 exhibited a culture medium-dependent inhibition on *M. oryzae*, with the stronger effect seen on LBA and PDA than on KBA (Figure 5A), suggesting a role of volatile compounds in suppressing the growth of *M. oryzae*. Some *Pseudomonas* species could synthesize hydrogen cyanide, a potent inhibitor of cytochrome *c* oxidase, under microaerobic conditions (O_2_ < 5%) for a better niche competition [14]. The emission of hydrogen cyanide by *B. gladioli* BBB-01 was examined by placing a carbonate-picrate paper strip in the headspace of sealed culture plates (Figure 5B). As expected, a *P. aeruginosa* strain, #6 isolate in this study, could produce hydrogen cyanide when it grew on LBA and KBA. By contrast, *B. gladioli* BBB-01 did not emit hydrogen cyanide under the culture condition.

### 2.4. Chemical Identification of VOCs Produced by B. gladioli BBB-01

Although VOCs released by a wide range of microorganisms have been documented in a large body of literature [15,16,17,18], few information is available for *B. gladioli*. In this study, VOCs produced by *B. gladioli* BBB-01 were collected by the SPME fiber and analyzed by gas chromatography-mass spectrometry (GC-MS). A peak with a retention time of 3.61 min in the GC chromatogram appeared in the sample collected from *B. gladioli* BBB-01 grown on LBA but not from blank LBA (Figure 6A). The chemical entity of the peak was further analyzed by mass spectrometry, and dimethyl disulfide was suggested based on the NIST reference database with a similarity of 95% (Figure 6B). On the other hand, a peak with a retention time of 2.86 min appeared in the collected sample if *B. gladioli* BBB-01 was cultivated on PDA (Figure 6C). The fragmentation pattern in the mass spectrum suggests that the compound is 2,5-dimethylfuran with a similarity of 97% (Figure 6D). The purchased dimethyl disulfide and 2,5-dimethylfuran were subjected to the GC analysis as the standards to confirm the identities of the volatile compounds emitted by *B. gladioli* BBB-01. Either of the standards had a nearly identical retention time, <2%, to the respective value described above.

### 2.5. Antifungal Activity of 2,5-Dimethylfuran and Dimethyl Disulfide via Fumigation

Dimethyl disulfide is a volatile compound emitted by phylogenetically diverse bacteria and fungi. Thanks to its pest control activity, this chemical has been used as an active ingredient in commercial fumigants, e.g., Paladin^®^. By contrast, to the best of our knowledge, 2,5-dimethylfuran produced by bacteria has not been reported in the literature. In this study, the activity of the vapor of 2,5-dimethylfuran and dimethyl disulfide against the growth of fungi was tested and the effects were compared with each other. The fungi tested include five plant pathogens including *M. oryza*, *G. fujikuroi*, *S. oryzae*, *P. noxius*, *Colletotrichum fructicola*, and one human pathogen, i.e., *Candida albicans*. The amounts of 2,5-dimethylfuran or dimethyl disulfide applied in a 125 mL Erlenmeyer flask were 1, 5, 10, 25, 50, and 100 µL. The radial inhibition percentage of the tested fungi by the two volatile compounds at each dosage is indicated in Figure 7. Both 2,5-dimethylfuran and dimethyl disulfide exhibited a dosage-dependent inhibition effect on all the tested fungi. In general, 2,5-dimethylfuran exhibited a more potent inhibition effect than dimethyl disulfide except in the case against *C. fructicola*. The vulnerability of the fungi under fumigation by 2,5-dimethylfuran was in the order of *M. oryza* > *P. noxius* > *G. fujikuroi* > *C. fructicola* > *S. oryzae* > *C. albicans*, and that by dimethyl disulfide was in the order of *C. fructicola* > *G. fujikuroi* > *P. noxius* > *M. oryza* > *S. oryzae* > *C. albicans*. The morphology of the suppressed *M. oryza* under sub-lethal concentrations of 2,5-dimethylfuran was also observed by scanning electron microscopy. Different from the observation of fragmented mycelia in confrontation plate, the look of the suppressed fungus was not changed apparently.

## 3. Discussion

Numerous beneficial bacteria, particularly in the genus *Bacillus* and *Pseudomonas*, have been reported to suppress the growth of plant pathogens, including bacteria, fungi, and nematodes, through antagonistic behavior. Recently, the genus *Burkholderia* is recognized also as a rich source for potential biocontrol agents in agriculture. [19]. Taking strains in the species *B. gladioli* as examples, *B. gladioli* B111 inhibited the occurrence of lily gray mold *Botrytis ellipticaon* [20], a *B. gladioli* pv. *agaricicola* strain controlled a wide range of fungi including *Botrytis cinerea*, *Aspergillus flavus*, *Aspergillus niger*, *Penicillium digitatum*, *Penicillium expansum*, *Sclerotinia sclerotiorum*, and *Phytophthora cactorum* [21], *B. gladioli* 3A12 suppressed *Sclerotinia homoeocarpa* [22], and *B. gladioli* NGJ1 inhibited *Rhizoctonia solani* via a rather unusual mycophagous behavior [23]. In this study, we demonstrated that *B. gladioli* BBB-01 could inhibit the growth of a variety of pathogenic fungi through releasing water-diffusible substances and emitting volatile compounds.

Several dozens of VOCs emitted by the genus *Burkholderia* have been identified. *B. gladioli* pv. *agaricicola* strain ICMP 11096 was able to produce d-limonene [21]. *B. ambifaria* emitted a blend of volatile compounds, which not only inhibited the growth of *Rhizoctonia solani* and *Alternaria alternate* but also induced significant biomass increase in *Arabidopsis thaliana* [24]. The *B. ambifaria*-emitted volatiles consisted of about 40 compounds, among which dimethyl disulfide was the most abundant. *Paraburkholderia tropica*, formerly known as *B. tropica*, emitted rich VOCs including acetic acid, methyl hexadecanoate, dimethyl disulfide, isobutyl ether, toluene, *p*-xylene, 5-cyano-1,2,3-thiadiazole, tetrachloroethyene, tricosene, 3-methoxybutyl-1-ene, methylcyclohexane, nonane, ethylbenzene, ethyl valerate, ocimene, α-pinene, d-limonene, and l-fenchone [25]. In this study, we found that *B. gladioli* BBB-01 could emit 2,5-dimethylfuran and dimethyl disulfide. Both are potent in growth suppression of *M. oryza*, *G. fujikuroi*, *S. oryzae*, *P. noxius*, *C. fructicola*, and *C. albicans* via fumigation. To our knowledge, this is the first report describing the emission of 2,5-dimethylfuran by a bacterial strain and demonstrating the efficacy of 2,5-dimethylfuran vapor in suppressing the growth of pathogenic fungi. From the application perspective, the finding in this study provides another alternative to control plant fungal disease by using 2,5-dimethylfuran as a fumigant.

The emitters of dimethyl disulfide include a wide range of fungi and bacteria. l-methionine γ-lyase is the conserved key enzyme for biosynthesis of dimethyl disulfide [26]. In fact, the coding gene of the lyase (*mdeA*) is located on the chromosome 2 of *B. gladioli* BBB-01. By contrast, the known emitters of 2,5-dimethylfuran is limited to a couple of fungi such as *Penicillium commune* and *Paecilomyces variotii* [9]. However, how 2,5-dimethylfuran is synthesized by microorganisms is still unknown. Through emitting 2,5-dimethylfuran, *B. gladioli* BBB-01 may gain an additional advantage over its rivals in the habitats. Microbial production of hydrogen cyanide relies on the *hcnABC* gene cluster [27]. Although *B. gladioli* BBB-01 contains *hcnABC* on the chromosome 2, this study failed to find evidence to support hydrogen cyanide production. Presumably, certain culture conditions, e.g., strictly controlled microaerobic environment, are required for *B. gladioli* BBB-01 to emit hydrogen cyanide.

Liquid diffusible substances such as antibiotics and fungal cell wall-hydrolytic enzymes might also play a role in the antagonistic activity of *B. gladioli* BBB-01 based on the results of confrontation plate assay (Figure 1A). Searching the genome of *B. gladioli* BBB-01 using the MIBiG database by antiSMASH discovered nine BGCs potentially able to direct the synthesis of pesticidal compounds. Of them, a BGC located on chromosome 1 shows 30% sequence similarity to BGC0000398, which is responsible for the production of cyclic lipopeptide orfamide B. This lipopeptide is a biosurfactant with fungicidal and insecticidal activity [28,29,30,31]. Therefore, *B. gladioli* BBB-01 may produce an orfamide analog to suppress the pathogenic fungi. Besides, there are two chitinase homologous genes in chromosome 2 of *B. gladioli* BBB-01. The secreted chitinase may damage the fungal cell wall, leading to fragmentation of the confronted mycelia. Nonetheless, the actual effective compounds released by *B. gladioli* BBB-01 in agar plate remain to be identified.

*B. gladioli* BBB-01 was isolated from rice shoot. Spraying *B. gladioli* BBB-01 on rice did not cause observable harm to the plant. The confrontation assay in this study showed an antagonistic activity of this bacterial strain against *M. oryzae*, *G. fujikuroi*, and *S. oryzae*, three common pathogenic fungi of rice in Taiwan. In addition, both dimethyl disulfide and 2,5-dimethylfuran have the power to control the growth of a variety of fungi. Thus, the potential of *B. gladioli* BBB-01 in protection of rice from infestation by pathogenic fungi *in planta* shall be assessed in the future. Queries such as how the fungal cells are suppressed by 2,5-dimethylfuran and what genes are involved in the biosynthesis of 2,5-dimethylfuran in *B. gladioli* BBB-01 are important issues that deserve further investigation.

## 4. Materials and Methods

### 4.1. Media and Chemicals

Potato infusion powder, peptone, tryptone, and yeast extract were purchased from Difco^TM^ (BD, Franklin Lakes, NJ, USA). Dimethyl disulfide was purchased from Alfa Aesar (Ward Hill, MA, USA), while 2,5-dimethylfuran was purchased from Acros Organics (Geel, Belgium). Other general chemicals were purchased from Sigma-Aldrich (St. Louis, MO, USA) and Merck (Darmstadt, Germany).

### 4.2. The Screen of Antifungal Bacteria

The fungus as indicated was cultivated on a PDA plate (0.4% potato infusion powder, 2% glucose, 1.5% agar) at 28 °C for 4 days. A 1-cm-diameter agar plug, full of mycelium, was cut out from the 4-day-cultured fungal colony and positioned onto the center of another plate of PDA, KBA (2% peptone, 0.15% K_2_HPO_4_, 0.15% MgSO_4_·7H_2_O, 1.5% glycerol, 1.5% agar, pH 7.2) or LBA (1% tryptone, 0.5% yeast extract, 1% NaCl, 1.5% agar). The fungus was cultivated continuously until the diameter of the radial colony reached about 6 cm in diameter. Subsequently, a 200-µL aliquot of the overnight culture of screened microorganisms was dropped onto a 0.6-cm-diameter filter paper, placed at a distance of 0.5 cm to the periphery of the fungal colony. The plate was incubated continuously at 28 °C, and the growth of the pathogenic fungus confronted by screened microorganisms was recorded daily. The incubator used in this study was made by Yih Der Technology Co., (New Taipei City, Taiwan).

### 4.3. Pangenomic Analysis of B. gladioli Isolates

In this study, the whole genome of *B*. *gladioli* BBB-01 was sequenced by performing 2 × 301 bp paired-end sequencing using Illumina MiSeq System (San Diego, CA, USA), followed by long-read sequencing using Nanopore technology (Oxford Nanopore Technologies, Oxford, UK). The complete circular chromosomes of BBB-01 were displayed using DNAplotter [32]. The nucleotide sequences of the chromosomes and plasmid are available in Genbank with accession number CP068049, CP068050, and CP068051, respectively. A genome mining for biosynthetic gene clusters (BGCs) that are responsible for the production of various secondary metabolites was conducted using antiSMASH version 5.2.0 [33]. For the kinship analysis of *B. gladioli* BBB-01, the sequencing reads of other 22 *B. gladioli* isolates, representing different clades of *B. gladioli* in the phylogenetic tree described by Jones et al. [10], were collected from the database (Table 1). The sequencing reads of each strain were individually *de novo* assembled using Unicycler (v0.4.8) [34]. The bacterial genomes were annotated by Prokka (v1.14.5) [11], followed by pangenome analysis using Roary (v3.11.2) [12]. Each gene in pangenome was assigned to Clusters of Orthologous Groups of proteins (COGs) using the Perl script cdd2cog.pl [35], and the distribution of genetic contents in pangenome was analyzed by heatmap clustering using the integrated tools in R software (version 4.0.2). The core genes in the pangenome were concatenated for multiple nucleotide sequence alignment using MAFFT (v. 7.455) [36]. Accordingly, a maximum-likelihood tree was constructed using RAxML (v. 8.2.12) in the GTR-GAMMA model with 1000 rapid bootstraps [12]. The phylogenetic tree was displayed using Molecular Evolutionary Genetics Analysis software (MAGA X) [37].

### 4.4. Assay of Bacterial Antifungal VOCs

A 100-µL aliquot of the overnight culture of *B. gladioli* BBB-01 was spread out on a PDA, LBA, or KBA plate, followed by incubation at 28 °C for 24 h. Meanwhile, *M. oryzae* was cultivated on a separate PDA plate until the colony size reached about 2 cm in diameter. The two agar plates, one with *B. gladioli* BBB-01 and the other with *M. oryzae*, were aligned face-to-face and sealed with parafilm. The sealed dish unit was incubated at 28 °C, and the growth of *M. oryzae* was recorded daily. In the control experiment, the plate of *M. oryzae* was sealed with a blank medium plate.

### 4.5. Assay of Bacterium-Emitted Hydrogen Cyanide

A 100-µL aliquot of overnight bacterial culture was spread out on the plate of PDA, LBA, or KBA, followed by incubation at 28 °C for 24 h. A cellulose strip (3 cm × 1 cm) (Whatman^®^, Kent, UK) that had been soaked in a solution of 0.5% picric acid and 2% sodium carbonate was stuck on the interior surface of the lid of the plate. The bacterium in the parafilm-sealed plate was grown continuously at 28 °C, in dark, for 4 days. A color change in the cellulose paper from yellow to reddish-brown indicates the presence of hydrogen cyanide.

### 4.6. Chemical Identification of Bacterial VOCs by GC-MS

A 100-µL aliquot of overnight bacterial culture was spread out on 5 mL PDA or LBA in a 125 mL Erlenmeyer flask. The flask mouth was sealed with several layers of parafilm and incubated at 28 °C for 3 days. Then, a solid-phase microextraction (SPME) needle (75 μm Car/PDMS, Supelco, Bellefonte, PA, USA) was used to pierce through the parafilm, and the needle was held for 20 min to adsorb the VOCs accumulated in the headspace of the flask. Adsorbed VOCs were analyzed by GC-MS using the TRACE GC PolarisQ mass system (Thermo Fisher Scientific, Waltham, MA, USA). The operation of GC was carried out using a DB-5ms column (30 m × 0.25 mm i.d., 025 film thickness) (Agilent J & W Scientific, Santa Clara, CA, USA) with a helium flow at 1 mL/min under the temperature program: 240 °C, 10 sec (desorption); 40 °C, 5 min; 40 → 120 °C at the accelerating rate of 3 °C/min; 120 → 180 °C at 4 °C/min; 180 → 280 °C at 20 °C/min; and 280 °C, 5 min. The compound representing each of the chromatographic peaks was then analyzed by MS. The chemical structure of the compound was proposed based on the fragmentation pattern against the database in the National Institute of Standards and Technology (NIST) by using the NIST 08 MS Library and MS Search Program v.2.0f.

### 4.7. Antifungal Activity Assay of Chemically Synthesized Fumigants

The vulnerability of 5 plant pathogens (*M. oryza*, *G. fujikuroi*, *S. oryzae*, *P. noxius*, and *Colletotrichum fructicola*) and 1 human pathogen (*Candida albicans*) under the fumigation were tested in this study. To plant pathogens, a 0.6-cm-diameter agar plug, full of the tested fungal mycelium, was positioned on the center of PDA molded in a 125-mL Erlenmeyer flask. To *C. albicans*, 5 µL overnight culture was deposited on the center of YPD agar (1% yeast extract, 2% peptone, 2% glucose, and 1.5% agar). The flask was placed upside down and an aliquot of 2,5-dimethylfuran or dimethyl disulfide as indicated was dropped onto a small cotton ball attached to the interior side of the screw cap of the flask. The cap was closed tightly, and the fungus inside was grown continuously for 1–2 weeks. The effect of the fumigant on growth of the fungus was recorded, and the radial inhibition percentages (%) was calculated as [(*Rc* − *Ri*)/*Rc*] × 100, in which *Rc* is the value of the fungal growth radius in the absence of fumigant and *Ri* represents the growth radius in the presence of fumigant. The data presented are the mean of three replicates.

### 4.8. Scanning Electron Microscopy (SEM)

*M. oryzae* was inoculated onto the center of a PDA plate until it grew up to a colony of 3 cm in diameter. A 200-µL aliquot of the overnight culture of *B. gladioli* BBB-01 or culture medium (the control) was added onto the PDA plate in a distance of 0.5 cm to the circumference of the mycelial colony, followed by incubation at 28 °C for 3 more days. A thin agar section was sliced off from the front line of the mycelial colony that directly confronted the *Burkholderia* strain or the control medium. The agar sample for scanning electron microscopy was prepared according to the instruction of the Cryo-SEM Preparation System (PP3010T, Quorum Technologies, Laughton, UK) and observed under the thermal fieldscanning electron microscope (JEOL, JSM-7800F, Tokyo, Japan).

## Figures and Tables

**Figure 1 molecules-26-00745-f001:**
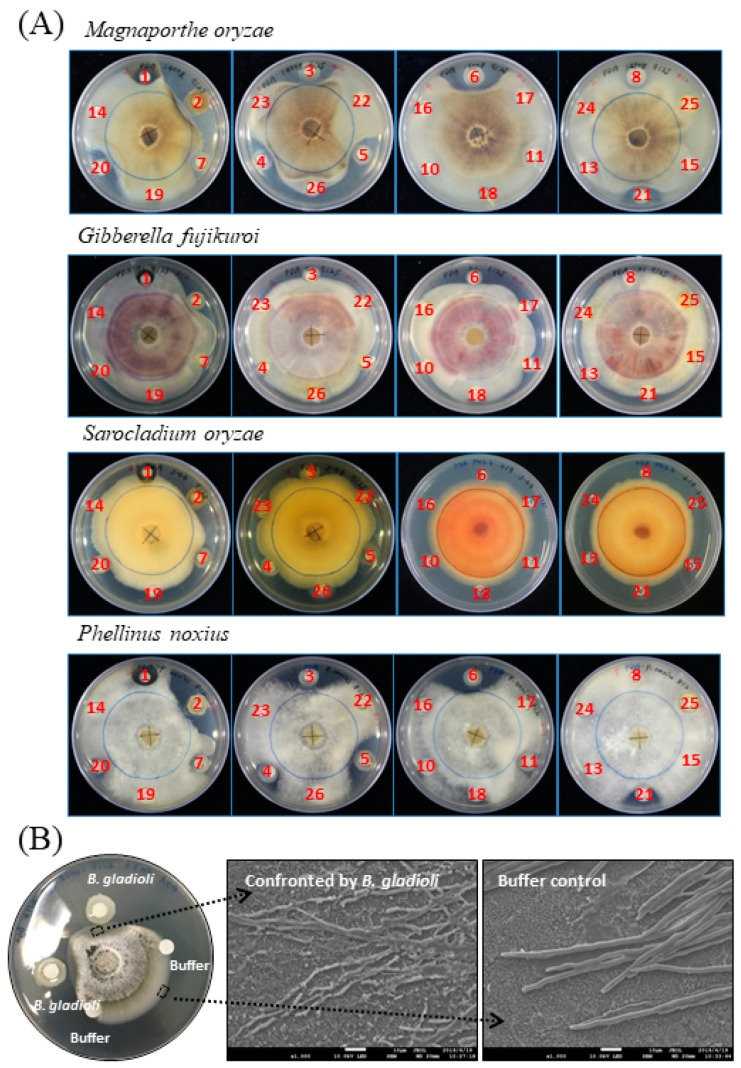
Confrontation assay between pathogenic fungi and screened microbial species. (**A**) The challenged fungus, as indicated, grew radially from the center of the plate, while the screened microbial species were placed outside and surround the mycelial colony as described in Section 4. The screened microbial species and the isolation site, in parentheses, are as follows: #1, *Aureobasidium melanogenum* (beehive); #2, *Burkholderia gladioli* (rice shoot); #3, *Pseudomonas aeruginosa* (spoiled food); #4, *P. aeruginosa* (spoiled food); #5, *P. aeruginosa* (mealworm); #6, *P. aeruginosa* (mealworm); #7, *Dyella yeojuensis* (urban soil); #8, *Serratia marcescens* (bamboo forest soil); #10, *P. aeruginosa* (rice field); #11, *P. aeruginosa* (rice field); #13, *S. marcescens* (rice field); #14, *Aeromonas* sp. (rice field); #15, *S. marcescens* (cigarette beetle); #16, *S. marcescens* (cigarette beetle); #17, *S. marcescens* (riverbank soil); #18, *S. marcescens* (riverbank soil); #19, *Bacillus* sp. (poultry feather); #20, *Burkholderia cepacia* (urban soil); #21, *P. aeruginosa* (urban soil); #22, *S. marcescens* (urban soil); #23, *P. aeruginosa* (urban soil); #24, *Stenotrophomonas nitrireducens* (used coffee grounds); #25, *Sphingobium yanoikuyae* (used coffee grounds). #26, *Pseudomonas* sp. (alkaline soil) (**B**) Morphology change of *M. oryzae* in response to the confrontation against *B. gladioli* BBB-01 was examined under a scanning electron microscope as described in Section 4. The mycelia within the dashed line-enclosed area were taken and observed. The scale bar in photos represents 10 µm.

**Figure 2 molecules-26-00745-f002:**
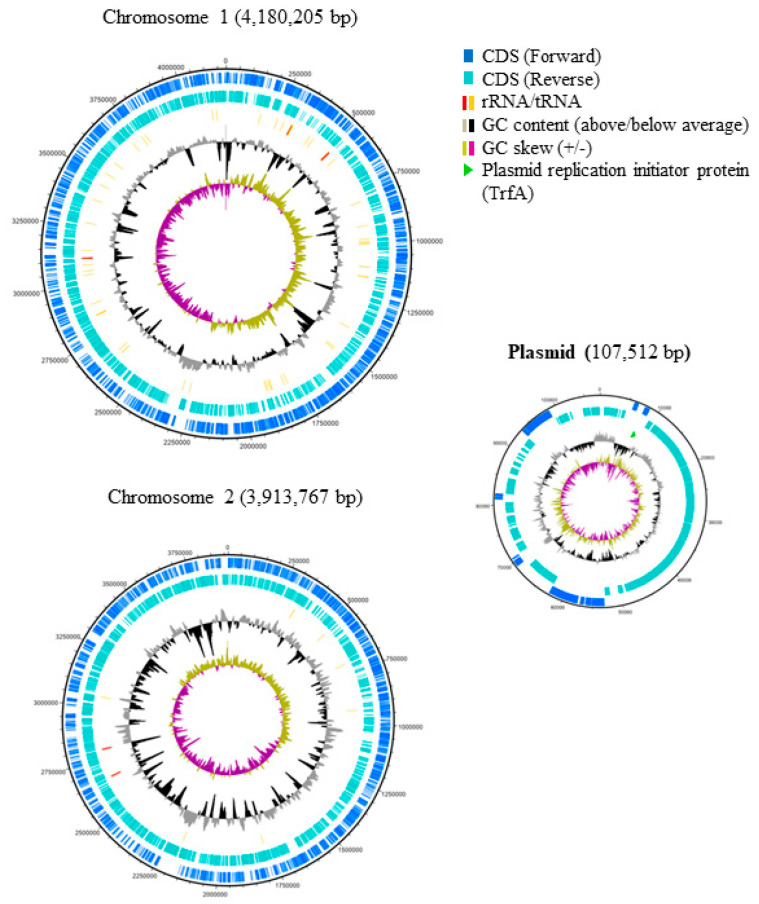
Genome maps of *B. gladioli* BBB-01. The nucleotide sequence of the genome was determined as described in Section 4. The genetic features were labeled from the outer to the inner rings of the maps. The 1st and 2nd rings indicate the protein-coding sequences (CDS) encoded in the forward and reverse strands of genome, respectively. The 3rd ring indicates the genes of rRNA and tRNA with red and yellow sticks, respectively. The 4th ring indicates GC content. Those being above and below the average are shown by grey and black curves, respectively. The 5th ring indicates GC skew. The positive and negative skew are shown by the light gold and purple curves, respectively. The light green arrowhead in the plasmid map indicates the replication initiator protein.

**Figure 3 molecules-26-00745-f003:**
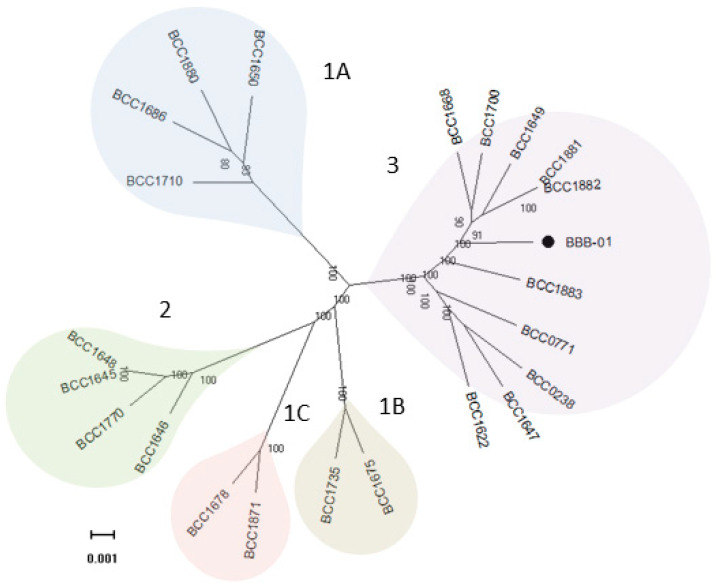
Phylogenetic relationship of 23 *B. gladioli* strains. The genomes of 22 previously reported *B. gladioli* strains and the strain isolated in this study (BBB-01) were analyzed and compared on the genome scale, from which 4141 core genes were identified. The concatenated core genes were aligned and used to construct a maximum-likelihood tree with 1000 rapid bootstraps. All the bioinformatics tools used in the construction of this phylogeny are described in Section 4. The scale bar represents the number of base substitutions per site.

**Figure 4 molecules-26-00745-f004:**
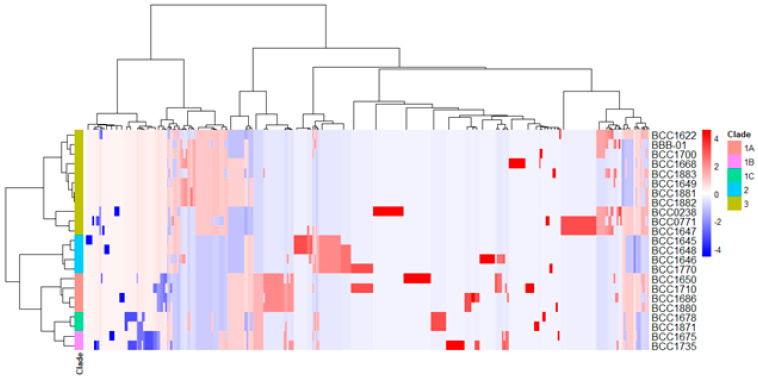
Heatmap clustering for the distribution of genes involved in secondary metabolite biosynthesis in *Burkholderia gladioli* strains. The 458 genes, which are involved in secondary metabolite biosynthesis (COG category Q) in the pangenome of 23 *B. gladioli* strains, were analyzed by heatmap clustering using the integrated tool (pheatmap) in the R program. The number of each gene among the 23 *B. gladioli* strains was normalized, followed by construction of distance matrix using the Manhattan method and hierarchical clustering using the Ward’s minimum variance (ward.D) method. The clade classification based on core genes is shown vertically, while the clustering of the 458 genes is shown horizontally.

**Figure 5 molecules-26-00745-f005:**
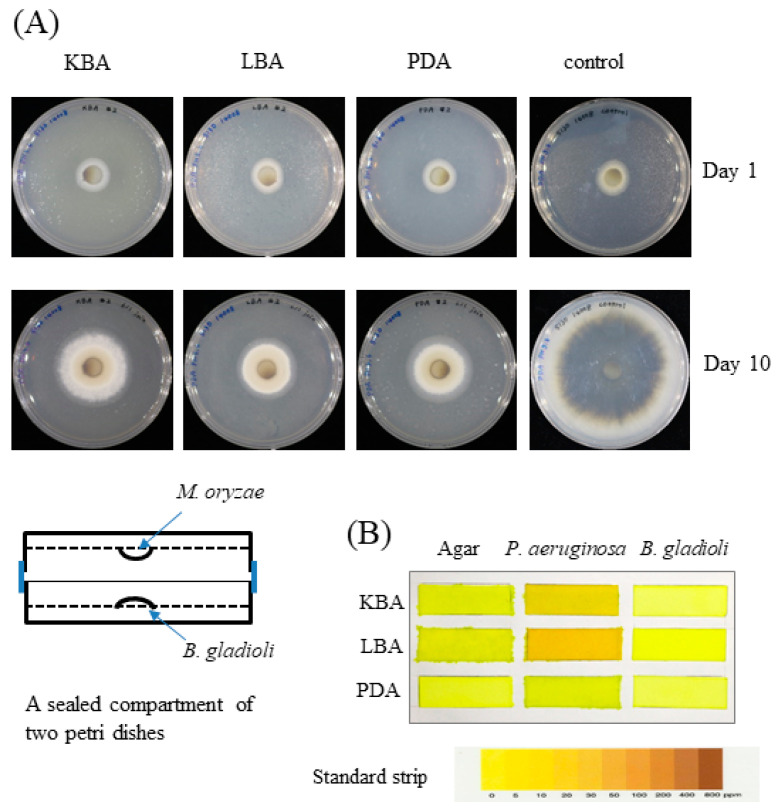
Inhibition of *M. oryzae* by *B. gladioli* BBB-01 via fumigation. (**A**) *M. oryzae* and *B. gladioli* BBB-01 were cultivated separately on two agar plates within a sealed compartment as illustrated. *M. oryzae* was grown on PDA, while *B. gladioli* BBB-01 was grown on KBA, LBA, or PDA as indicated. The growth of *M. oryzae* alone was taken as the control. The radial colony of *M. oryzae* on days 0 and 10 after fumigation was shown. (**B**) *P. aeruginosa* and *B. gladioli* BBB-01 were cultivated on KBA, LBA, or PDA. A paper strip soaked with picric acid and sodium carbonate was stuck on the lid within the sealed petri dish. The appearance of brown color of the paper indicates the presence of hydrogen cyanide as indicated by the standard strip.

**Figure 6 molecules-26-00745-f006:**
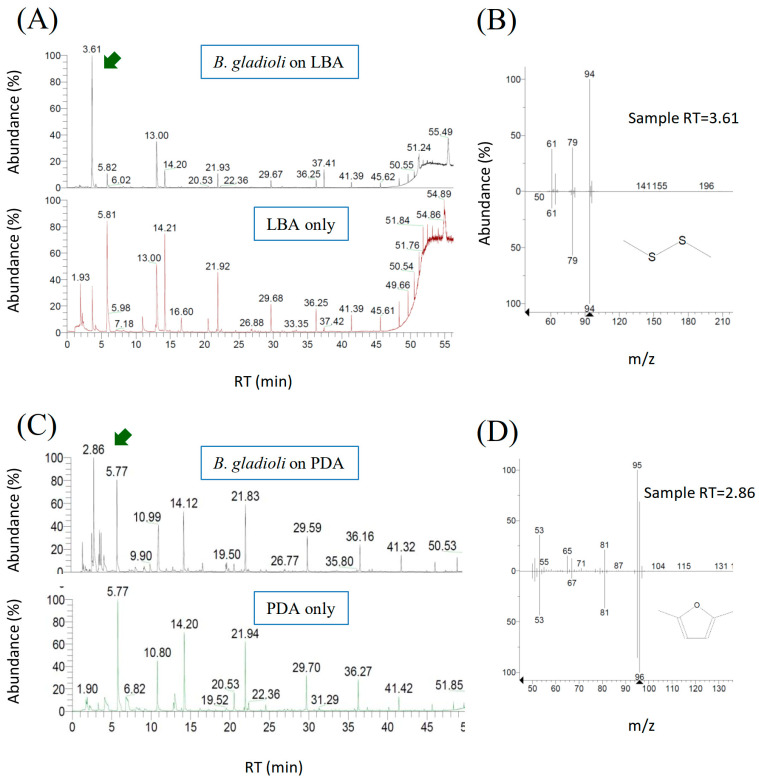
Chemical identification of VOCs emitted by *B. gladioli* BBB-01. VOCs collected from the headspace of the bacterial culture flask by the SPME fiber was analyzed by GC (**A**,**C**). The peaks of a retention time (RT) of 3.61 and 2.86 min appeared only when the *Burkholderia* strain was grown on LBA and PDA, respectively. The two unique peaks were further analyzed by mass spectrometry, and the fragmentation patterns match that of dimethyl disulfide (**B**) and 2,5-dimethylfuran (**D**), respectively, according to the NIST database.

**Figure 7 molecules-26-00745-f007:**
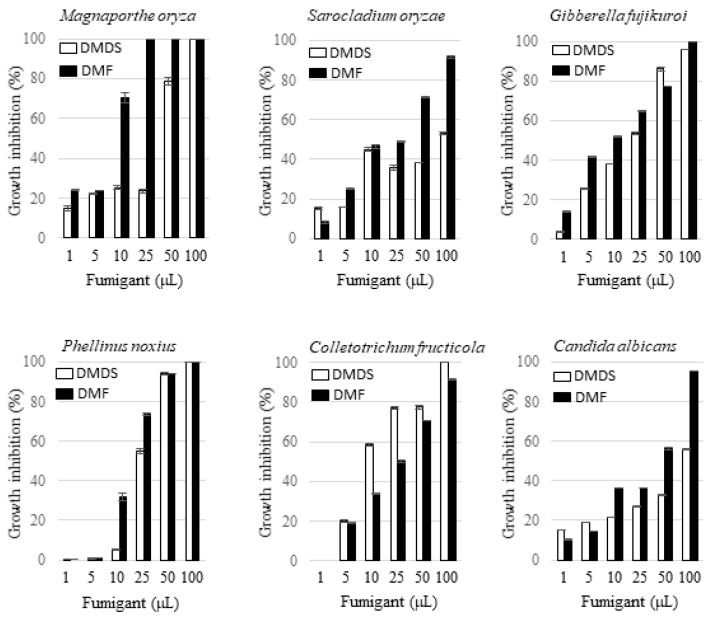
Antifungal activity of 2,5-dimethylfuran and dimethyl disulfide via fumigation. The fungus was cultivated on PDA or YPD in a tightly closed 125 mL Erlenmeyer flask, in which 2,5-dimethylfuran (DMF) or dimethyl disulfide (DMDS) at the indicated aliquot was included as described in Section 4. Cultivation was continued until the fungal colony in the fumigant-absent control group was fully grown. The effect of the fumigant on growth of the fungus was recorded, and the radial inhibition percentages (%) was calculated as described in Section 4. All data points are the means of three replicates.

**Table 1 molecules-26-00745-t001:** *Burkholderia gladioli* isolates selected in the pangenome analysis in this study.

Clade	Strain Name	Accession Number ^1^	Synonym	Bongkrekic Acid Production	Remark
1A	BCC1710	ERS785039		+	CF isolate ^2^
	BCC1650	ERS784828	ENV; *B. gladioli* pv *cocovenenans*, LMG 11626, Indonesia	+	
	BCC1880	ERS1371628	ENV; *B. gladioli* pv *cocovenenans*, LMG 18113, China	+	
	BCC1686	ERS784816		+	CF isolate
1B	BCC1675	ERS785062		+	CF isolate
	BCC1735	ERS785008		+	CF isolate
1C	BCC1678	ERS784880		+	CF isolate
	BCC1871	ERS1371626			CF isolate
2	BCC1645	ERS784911	ENV; *B. gladioli* pv. *alliicola*, LMG 6954, Australia		
	BCC1648	ERS784957	ENV; *B. gladioli* pv. *alliicola*, LMG 2121, USA		
	BCC1770	ERS1328772			CF isolate
	BCC1646	ERS784928	ENV; *B. gladioli* pv. *alliicola*, LMG 6878, India		
3	BCC1622	ERS784864			CF isolate
	BCC1647	ERS784943	*B. gladioli* pv. *gladioli*, LMG 6882, USA		
	BCC0238	ERS784907			CF isolate
	BCC0771	ERS784806	*B. gladioli* pv. *gladioli*, LMG 2216 (ATCC 10248), USA		
	BCC1668	ERS784974			CF isolate
	BCC1700	ERS784851			CF isolate
	BCC1649	ERS784812	ENV; *B. gladioli* pv. *gladioli*, LMG 6880t4, Zimbabwe		
	BCC1881	ERS1371629	ENV; *B. gladioli* pv. *agaricicola*, NCPPB 3580, UK		
	BCC1882	ERS1371630	ENV; *B. gladioli* pv. *agaricicola*, NCPPB 3632, UK		
	BCC1883	ERS1371630	ENV; *B. gladioli* pv. *agaricicola*, NCPPB 3852, New Zealand		
	BBB-01		*B. gladioli* BBB-01		Isolated in this study

^1^ The genome is accessible at website https://www.ebi.ac.uk/ena/browser/home. ^2^ Isolates from cystic fibrosis patients.

## Data Availability

Not applicable.

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
