# Peer review of "Fungicidal Activity of Volatile Organic Compounds Emitted by Burkholderia gladioli Strain BBB-01"

_molecules, 2021, doi:10.3390/molecules26030745_

Round 1

Reviewer 1 Report

This is a good work on the effects of Burkholderia gladioli on different fungi, and the results are interesting.

I thing that the work is well described in the paper.

I only have two small considerations:

1) The word "bacteria" must appear in the abstract and in the Introduction. In the present form, the reader have to suppose that Burkholderia gladioli is a bacteria. Also, a small description of that bacteria must be included in the Introduction.

2) According to the text, for the identification of compounds you only use the fragmentation pattern against the MS libraries. Did you check the retention times against the synthetic standards that you have used? This is some times necessary to be sure of the identification. At least for the two interesting compounds (2,5-dimethylfuran and dimethyl disulfide).

Reviewer 2 Report

A paper entitled “Fungicidal activity of volatile organic compounds emitted by 2 Burkholderia gladioli strain BBB-01” is submitted to Molecules for further reviewing and publication. This research work provided a fast and simple way to evaluate the activity of volatile organic compounds. The paper is well written and very interesting to readers. I recommended that this paper is acceptable for publication with its present form.

Minor issues:

The authors indicate that they acquire the commercial standards but at no point do they indicate that they have verified the identity of the compounds by comparison with these standards.

On the other hand, it is necessary that they indicate the degree of similarity of the compounds with the standards listed in the NIST library.

Reviewer 3 Report

Dear Authors, 

congratulations on this very interesting manuscript.

I have only some minor comments to your work:

  1. in the 'materials and methods section' please, indicate the city and country of every producer of the reagents/materials/equipment used in the study
  2. add all used chemicals/reagents to the first subchapter of 'Materials and methods' section or indicate the producers later in the subchapters of 'Materials and methods'
  3. indicate the producer (city, country), the model of incubator used in the study, similarly to other equipment and materials
  4. please, prepare a table with the identification of volatile components by GC-MS. Please provide a list of retention indices to confirm the identity of the compounds
